# Identification of *DREB* Family Genes in Banana and Their Function under Drought and Cold Stress

**DOI:** 10.3390/plants13152119

**Published:** 2024-07-31

**Authors:** Yi Xu, Yanshu Zhang, Funing Ma, Jingxi Zhao, Huiting Yang, Shun Song, Shaoling Zhang

**Affiliations:** 1State Key Laboratory of Crop Genetics and Germplasm Enhancement, Sanya Institute of Nanjing Agricultural University, College of Horticulture, Nanjing Agricultural University, Nanjing 210095, China; xuyi@catas.cn (Y.X.); zhangyanshu@stu.njau.edu.cn (Y.Z.); 2State Key Laboratory of Biological Breeding for Tropical Crops, Tropical Crops Genetic Resources Institute, Chinese Academy of Tropical Agricultural Sciences, Haikou 571101, China; mafuning@catas.cn (F.M.); zhaojingxi2024@ynau.edu.cn (J.Z.); yanghuiting202404@ynau.edu.cn (H.Y.); 3Laboratory of Crop Gene Resources and Germplasm Enhancement in Southern China, Ministry of Agriculture and Rual Affairs, Key Laboratory of Tropical Crops Germplasm Resources Genetic Improvement and Innovation of Hainan Province, Haikou 571101, China; 4Hainan Seed Industry Laboratory, Sanya 572000, China; 5Hainan Key Laboratory for Biosafety Monitoring and Molecular Breeding in Off-Season Reproduction Regions, Sanya Research Institute, Chinese Academy of Tropical Agricultural Sciences, Sanya 572000, China

**Keywords:** genome-wide, DREB, climate change, drought, cold

## Abstract

Bananas are one of the most important cash crops in the tropics and subtropics. Drought and low-temperature stress affect the growth of banana. The *DREB* (dehydration responsive element binding protein) gene family, as one of the major transcription factor families, plays crucial roles in defense against abiotic stress. Currently, systematic analyses of the banana *DREB* (*MaDREB*) gene family have not yet been reported. In this study, 103 members of the *MaDREB* gene family were identified in the banana genome. In addition, transcriptomic analysis results revealed that *MaDREBs* responded to drought and cold stress. The expression of *MaDREB14/22/51* was induced by drought and cold stress; these geneswere selected for further analysis. The qRT-PCR validation results confirmed the transcriptome results. Additionally, transgenic Arabidopsis plants overexpressing *MaDREB14/22/51* exhibited enhanced resistance to drought and cold stress by reducing MDA content and increasing PRO and soluble sugar content. This study enhances our understanding of the function of the *MaDREB* gene family, provides new insights into their regulatory role under abiotic stress, and lays a good foundation for improving drought and cold stress-tolerant banana verities.

## 1. Introduction

Abiotic stresses such as drought and cold temperature limit plant growth and development, particularly under climate change circumstances [1]. As a result, plants have a variety of regulatory systems to protect themselves from the damage caused by harsh environmental conditions. Banana (*Musa acuminata* L.) is the fourth largest food crop in the world and is widely cultivated in tropical and subtropical regions. This majestic annual monocotyledonous herbaceous plant holds a special place in the hearts and diets of people worldwide. Its delectable fresh fruits are not only a delight to consume, but also play a crucial role in global food security and nutrition. The banana plant’s unique growth habits, nutritional value, and economic importance make it a cherished and indispensable crop in many tropical and subtropical regions. Banana cultivation requires sufficient water, as drought stress has a severe impact on banana plant growth and fruit development. Additionally, bananas are also highly sensitive to cold temperatures, with the critical temperature for growth inhibition set at 10 °C. A decrease to 5 °C can cause the leaves to turn yellow due to cold damage, while a further drop to 2.5 °C may lead to injury, resulting in the rotting and death of the pseudostem center. Therefore, abiotic stresses, such as drought and temperature, adversely affect both growth and productivity. Transcription factors are proteins that regulate gene expression by binding to specific cis-acting elements in gene promoters and play a crucial role in mediating plant responses to abiotic stress [2,3]. Several types of stress response transcription factors, including *MYB* [4], *WRKY* [5], *bZIP* [3,6] and *AP2/ERF* [7], are activated in response to abiotic stimuli. The *AP2*/*ERF* (*APETALA2*/ethylene-responsive factor) family consists of four major subfamilies: AP2, ERF, DREB and *RAV* [8,9]. Among these, the *DREB* subgroup is characterized by an AP2 domain that interacts with the DRE (Dehydration-Response Element) element [10,11,12]. The sequence A/GCCGAC found in genes responsive to drought and cold, is crucial for this interaction [13,14,15,16]. Based on the sequence similarities within their DNA-binding domains, DREB proteins are further categorized into six subgroups (A1–A6). Members of these subgroups serve distinct functions in plants. Notably, the A-1 subgroup, comprising *DREB1/CBF* (C-repeat Binding Factor)-like genes, is primarily regulated by low temperatures and orchestrates the expression of numerous cold stress-responsive genes. In contrast, the A-2 subgroup, consisting of *DREB2* genes, primarily functions in response to osmotic stress [8]. In mung bean (*Vigna radiata L.*), VrDREB2A is induced by drought and salt stresses, and its heterologous expression enhances tolerance to both drought and high salt stresses without compromising growth [17]. Similarly, in Arabidopsis, *DREB2A* and *DREB2B* actively respond to dehydration and high salinity stress [18]. Additionally, *BjDREB2* plays a pivotal role in ABA-independent gene expression during drought stress in Arabidopsis [19].

In recent years, *DREB* has been implicated in the response of plants to a wide range of abiotic stress. For example, 41 *DREB* genes have been identified in barley with most responding to drought and salt stress [20]. Overexpressing of *OsDREB1A* in transgenic plants has been shown to confer resistance to salt, drought, and low-temperature stress [21]. Similarly, the expression of *PpDBF1* is up-regulated during salt and drought stress [22], while *SbDREB2A* is induced under drought, NaCl and high-temperature stress [23]. Additionally, *AhDREB1* expression is induced under salt stress and Arabidopsis DREB1/CBF regulate the expression of many genes in response to drought and cold stress [24,25].

Several DREB transcription factor genes have been identified in various plants, including maize, Arabidopsis, soybean, and sesame [26,27]. However, there are fewer reports on this family in bananas. The sequencing of the entire banana genome has enabled a comprehensive examination of its genetic traits. As *DREBs* play a critical role in the response to various abiotic stresses, we selected this family for extensive study in banana. In this research, the *DREB* genes of banana (*MaDREB*) have been identified, also the phylogenetic relationship, gene structure, andexpression patterns have been analyzed. Specifically, three of the *MaDREBs*, *MaDREB14*, *MaDREB22*, and *MaDREB51*, were selected and transformed into Arabidopsis. The results provide a foundation for understanding the regulatory roles of *MaDREBs* and offer significant insights into banana bio-breeding and stress management.

## 2. Results

### 2.1. Phylogenetic Analysis

A total of 103 banana *MaDREBs* were identified, with lengths ranging from 447 bp to 1314 bp (Appendix A). The DREB family sequences of Arabidopsis were obtained from the TAIR website, and a tree was constructed to examine the relationship between the banana and Arabidopsis DREB family members. This tree was constructed by using the neighbor-joining method in MEGA-X (Figure 1). Based on the classification of Arabidopsis *DREB* families and the closeness of their sequence similarities, the banana *DREB* family was clustered into six subgroups: A1, A2, A3, A4, A5, and A6. These subgroups contained 8, 13, 2, 43, 25, and 12 genes, respectively.

Among these genes, some showed one-to-one correspondence with Arabidopsis *DREB* genes. For example, *MaDREB12* in the A4 subgroup is homologous to *AT4G16750*. Additionally, some of the genes have duplicated counterpart. For instance, in the A2 subgroup, *AT1G75490* and *AT3G57600* of correspond to *MaDREB17/98* and *MaDREB24/59*, respectively. In the A3 subgroup, *AT2G40220* corresponds to *MaDREB20/73*, and the *AT2G35700* gene of the A4 subgroup corresponds to *MaDREB12*.

### 2.2. Analysis of the Structural Domain

The similar motifs of the 103 members of the banana *DREB* family were predicted using the MEME website (Figure 2). The results showed that all identified members of the banana *DREB* family were structurally similar, containing motif1, motif2, and motif3. The gene structures of the identified banana *DREB* family members were visualized using TBtools, with the GFF3 annotation file as a reference. The results indicated that most banana *DREB* family members do not contain introns, only a few of the members contain 1–3 introns within their structures. These introns appear only as coding sequences (CDSs). Additionally, most members contain 5′ and 3′ untranslated regions (UTRs).

### 2.3. Analysis of MaDREB Promoters Structures

Promoter analysis of the 103 *MaDREB* genes (−2000 bp) revealed the presence of TATA and CATA motifs in all promoters. Additionally, a large number of cis-acting elements associated with abiotic stress responses were identified, including elements linked to salicylic acid, low temperature, abscisic acid, and light response. The promoters also contained hormonal response elements such as those responsive to auxin, MeJA (methyl jasmonate), and gibberellin (Figure 3).

### 2.4. Collinearity Analysis of MaDREBs

Gene duplication events are critical for plant genomes. To determine the duplication events of the *MaDREB* genes, we used TBtools to perform intraspecific covariance of banana *DREB* genes. Using Gene Density Profile and Gene Location Visualize from the GTF/GFF programs, the 103 identified *DREB* genes were mapped onto the chromosomes of the banana genome based on the genome annotation information (Figure 3). The DREB family genes are distributed on across ten chromosomes of the banana genome, with chromosome 10 having the highest number of genes (16), followed by chromosomes 2, 4, 5, 6, 9, and 11, each with containing nine genes. Chromosomes 4, 7, and 8 have 14, 12, and 7 genes, respectively.

All genes were displayed in a circle representation of the ten chromosomes, and 42 collinearity pairs were identified (Figure 4A). In the interspecific covariance analysis (Figure 4B), 1744 and 10,371 genes in Arabidopsis and rice, respectively, were found to collinear with genes in banana. Most of the DREB members in banana shared one or two homologous genes with Arabidopsis or rice.

### 2.5. Protein Interaction Analysis

An analysis of protein interactions revealed that the majority of the proteins interacting with *MaDREBs* were mainly AP2 (APETALA 2), ABF2 (ABRE binding factor 2), RHL41 (zinc finger protein 41), PFT1 (pore-forming toxin-like), HSFA3/A2 (heat shock transcription factor A3/A2), EXO (exocyst gene), and others. These proteins are mainly involved in plant resistance and development, providing important insights for further practical research into their cooperative roles and guiding principles (Figure 5).

### 2.6. Three MaDREBs Were Selected to Respond to Drought and Cold Stress

Based on the transcriptome data, there are many members that respond to drought and cold stress (Figure 6A). Notably, more members were found to respond to cold stress compared to drought stress. The members that responded to both stress conditions include *MaDREB14/22/25/36/51/67/79/80/81*. Among these, *MaDREB14*, *MaDREB22* and *MaDREB51* exhibited relatively high expression levels and were selected for further investigation. Further quantitative real-time PCR (qRT-PCR) analysis was subsequently conducted on these three genes (Figure 6B). The qRT-PCR results demonstrated that *MaDREB14*, *MaDREB22*, and *MaDREB51* were indeed responsive to both drought and cold stress, showing expression patterns consistent with the transcriptome findings.

### 2.7. Overexpression of MaDREB Enhances Transgenic Arabidopsis Tolerance to Drought and Cold Stress

The ability of DREBs to respond to abiotic stress in plants has been extensively studied. In this research, we focused on *MaDREB14/22/51*, which are induced under drought and cold stress (Figure 7).

The three genes were transformed into *Arabidopsis thaliana*, resulting in three independent transgenic lines for each gene. These transgenic lines were subjected to both drought and cold stress conditions. The results showed that under drought stress, all three genes enhanced the drought tolerance of the transgenic Arabidopsis plants, with *MaDREB14* showing the greatest improvement in drought resistance. In contrast, the phenotypic differences between transgenic plants and wild-type (WT) plants under cold stress were not significant. Nevertheless, upon returning to normal temperature, some of the phenotypic traits of the transgenic plants were restored.

### 2.8. Chlorophyll Fluorescence Measurements under Drought and Cold Stress

Chlorophyll fluorescence measurements were used to evaluate plant damage induced by drought and cold stress. Before stress treatments, there were minimal differences in chlorophyll fluorescence color between wild-type (WT) and transgenic plants However, under drought and cold stress, all plants were damaged to varying degrees, with transgenic plants being less damaged than WT (the false color of normal plant leaves was blue, whereas the false color of leaves changes from green to yellow or even black, indicating more damage to the plant). Among the transgenic lines, *MaDREB14* showed the least damage, as indicated by the fluorescence images (Figure 7).

### 2.9. Measurement of Physiological Indicators in Transgenic Arabidopsis

The expression levels of different transgenic *Arabidopsis thaliana* under drought and low-temperature stress were significantly higher than those under normal conditions (Figure 8). Physiological indicators, including malondialdehyde (MDA), proline (PRO), and soluble sugars, were measured in WT and transgenic Arabidopsis plants. Under normal conditions, there were no significant differences in MDA, PRO, and soluble sugar levels between WT and transgenic plants. However, after exposure to drought and cold stress, transgenic plants showed significantly lower MDA levels and higher PRO and soluble sugar levels compared to WT plants. These results indicate that *MaDREB14/22/51* enhance drought and cold tolerance in transgenic Arabidopsis (Figure 9).

## 3. Materials and Methods

### 3.1. Identification and Gene Family Analysis of DREBs in Banana

The protein sequence of the *M. acuminata* genome (DH-Phang-V4) was obtained from the Banana Genome Database (https://banana-genome-hub.southgreen.fr/) (accessed on 12 June 2023). Various analyses were conducted on the *MaDREB* gene family, including promoter element analysis, collinearity assessment and protein interaction prediction [28,29,30,31]. The relevant software and websites used for these analyses which are listed in Appendix A. Information on the CDS length, the protein length, the molecular formula, the molecular weight (MW) and the protein iso-electric point (pI) is listed in Appendix A.

### 3.2. Plant Materials, Transcriptome, and qRT-PCR Analysis

Differentially expressed genes were screened using banana drought and cold stress transcriptome data [32]. The raw data was deposited in NCBI-SRA database (accession number: PRJNA343716). The transcriptome data of MaDREBs under the drought and cold stress treatment are shown in Appendix A. Plants of the banana (*Musa acuminate* L. AAA group cv. Cavendish, BX) were used in this study. The seedlings (five-leaf stage)were planted in an incubator with 28 °C with a 16 h light/8 h dark cycle and 70% relative humidity. The stress treatment was applied to banana plants at the five-leaf stage. For the cold stress, plants were treated for 22 h at 4 °C. For the drought treatment, plants were treated with 200 mM mannitol for 7 days at 28 °C. These samples were used for the qRT-PCR analysis, which were using the instruments [31]. Relative expression was determined by 2-∆∆Ct and normalized to MaDREBs.

### 3.3. Cloning and Vector Construction of MaDREB14/22/51

The cDNA of *MaDREB14/22/51* and its promoter region were amplified by PCR from the banana (*Musa acuminate* L. AAA group cv. Cavendish, BX) and then cloned separately into the pCAMBIA1304 vector, named pCAMBIA1304-*MaDREB14/22/51*. The oligo primers of *MaDREBs* used for qRT-PCR are shown in Appendix A.

### 3.4. Plant Transformation and Drought and Cold Treatment

Agrobacterium transformed with pCAMBIA1304-*MaDREB14/22/51* was cultured overnight with shaking. Gene transformation experiments were conducted using flowering Arabidopsis thaliana via the pollen tube passage method [33]. The transgenic Arabidopsis (30 d) of the T3 generation were subjected to drought and cold treatments. For the drought and cold stress, the plants were treated in the incubator, respectively [34]. Three independent lines were taken from each transgenic plant for the experiment.

### 3.5. Analyses of Physiological Indices

The content of Malondialdehyde (MDA), proline and soluble sugar were determined using the detection kits (SH113W-100, Jinkelong, Beijing, China; SH140K-50, Jinkelong, Beijing, China; A145-1-1, Jiancheng, Nanjing, China). Chlorophyll fluorescence images were taken with a camera mounted above the plant pot. The background was removed from the image and only the plant was photographed. The camera has a 10 × 13 cm field of view and a resolution of 640 × 480 pixels. Light-emitting diodes (LEDs) were placed around the lens of the camera. The fluorescence intensity is displayed in false colours. A 450 nm blue LED provides pulse modulated excitation light for both light and saturation pulses, and a red long pass filter in front of the CCD chip limits the wavelength to 620 nm. Wild type and four plants from each transgenic line were tested separately.

## 4. Discussion

Many *DREB* genes have been cloned as more gene families from various species are mined and analyzed. Research has shown that *DREBs* play a crucial role in regulating gene expression in response to abiotic stresses such as drought and cold. However, the characteristics and functions of banana *DREBs* remains unclear. In this study, *DREB* family members were identified in the banana genome and the potential roles of *MaDREBs* in responses to abiotic stress were elucidated.

Recently, several *DREB* homologues have been identified in various plants including soybean [35], Arabidopsis [36], wheat [37], sorghum [38], rice [39], tomato [40], tobacco [41], barley [42], grape, maize [43], and Chinese cabbage [44]. The numbers of *DREB* gene members vary among some species. This study identified 103 *DREB* genes in the banana, a number similar to the 172 *DREB* genes found in alfalfa [45]. In comparison, the numbers of *DREB* genes in maize [46], grape, Arabidopsis, rice, and Chinese cabbage are 18, 38, 56, 56, and 65, respectively [44]. Additionally, cotton contains a significantly higher number of *DREBs* (totaling 535).

In this research, the *MaDREBs* were subdivided into six groups based on the classification in *Arabidopsis thaliana*, a result consistent with that of *Brassica rapa* [44] and cotton [47]. In contrast, *Medicago sativa* has only five groups [45]. The *MaDREB* members are distributed on chromosomes 2–11 with a total of 103 members, exhibiting collinearity among them. This result is similar to findings in in *B. oleracea* [48]. Elemental analysis further elucidates the structural unity of gene family members. In Figure 2, the banana *DREB* members contain three homologous elements that are neatly arranged. However, the number of elements in the gene sequences is not consistent across species, with ten elements present in the sequences of the *MsDREB* members [45].

Introns are sequences of nucleotides in the coding region of a gene that do not code for a protein and areimportant parts of the gene’s structure. In contrast, exons are nucleotide sequences that are responsible for translating proteins, and the most critical part of the gene. The exon and intron structure can reveal links within gene families [49]. In banana, most *MaDREB* family members do not contain introns; only 34 out of 103 DREB members have introns, and most genes contain only one exon. In *MsDREB*, the number of exons vary from one to four. For example, the ‘*gene62759*’ contains one intron and four exons. This indicates significant variation in the number and content of motifs between different subfamilies of *MsDREBs*, with their less variable gene structures, where 137 members have only one exon or no intron. Gene structure analysis showed that introns are absent in most *FvDREBs* [20,50]. Meanwhile, all FvDREB protein sequences contain elements associated with the AP2 domain, indicating that the AP2 domain is highly homologous in *FvDREBs*. Similarly, most of the *DREB* member genes in cotton do not contain introns. The proportions of UTRs and CDSs also differed among the four cotton species. The differences in the structure of the *DREB* genes indicate a diversity of functions [51].

Cis-acting elements are specific DNA sequences with transcriptional regulatory functions, which play an important role in gene transcription and expression [52]. The findings in this study are consistent with previous research on *BrDREB* in Brassica and *MnDREB* in mulberry [53]; our results suggest that the cis-acting elements of the *MaDREB* gene include elements that respond to abiotic stresses and hormones [44]. Furthermore, it has been demonstrated that DREB transcription factors are involved in the expression resistance genes through both ABA-dependent and ABA-independent pathways [54]. An ABA-related element was identified in the *MaDREB* sequence, suggesting that this gene may regulate downstream gene expression through an ABA-dependent signaling pathway. For instance, overexpression of *SlDREB3* in tomato has been shown to reduce ABA levels, thereby enhancing root growth and increasing photosynthetic rate [55]. Furthermore, the promoter region of the *DREB* gene contains a high number of light-responsive elements (Figure 5), indicating a potential link to circadian regulation. These results suggest that, in addition to the basic TATA-box and CAAT-box elements, the cis-acting elements in the *MaDREB* promoter can be categorized into three groups: stress-responsive, phytohormone-responsive, and growth and developmental elements. Abiotic stresses such as drought and cold temperatures adversely affect the plant growth and development [56]. It can lead to dehydration of plant cells and trigger a series of tolerance responses in plants [57]. As an important branch of *AP2*/*ERF*, many studies have been reported showing that *DREB* family members are valuable for improving crop stress tolerance for the past few years [48].

In this research, the results indicated that some *MaDREB* members responded to drought and cold stress, with *MaDREB1* and *MaDREB5* notably improving drought and cold-temperature tolerance in Arabidopsis. Similar findings have been reported in other studies. The *FvDREB* gene was shown to enhance drought tolerance in plants and was demonstrated by transcriptome sequencing and qRT-PCR assays [58]. *SsDREB2D*/*2F* improves drought and cold tolerance in sugarcane [59]. Beyond drought and cold stress, there are several reports showing that *DREB* genes respond to other abiotic stress. *BrDREB2B* was able to improve the survival of transgenic plants, which grew better under high-salt and high-temperature stress. Overexpression of *OsDREB2A* [60], *TaDREB3* [12], and *GmDREB2A* [61] in Arabidopsis has enhanced tolerance to various abiotic stresses [44]. Similarly, *ZmERF135* was significantly up-regulated after heat treatment, and the expression levels of its homologous genes, *OsDREB2A* and *AtDREB2A*, were previously reported to be similar [62]. Furthermore, overexpression of *TaDREB3* improved salt tolerance in wheat [12], a result similar to the expression of *DREB* in barley [42], *GmDREB2* in soybean, and *DvDREB2A* in chrysanthemum [63].

## 5. Conclusions

The *DREB* gene family plays a crucial role in enhancing plant stress tolerance. In this study, 103 *DREB* members were identified in banana. *MaDREBs* could respond to drought and cold stress. Among these, *MaDREB14*, *MaDREB22*, and *MaDREB51*, were highly responsive to drought and cold stress, and their expression levels were confirmed by qRT-PCR. The transgenic Arabidopsis with three genes showed drought and low-temperature tolerance. This study provided a new basis for understanding the regulatory functions of *MaDREBs* during abiotic stress and screened three stress-resistant genes as candidate members, laying a good foundation for further creation of stress-resistant germplasm. In the future work, we will conduct more in-depth studies to resolve the mechanism of action of *MaDREBs* to enhance drought and cold tolerance in plants.

## Figures and Tables

**Figure 1 plants-13-02119-f001:**
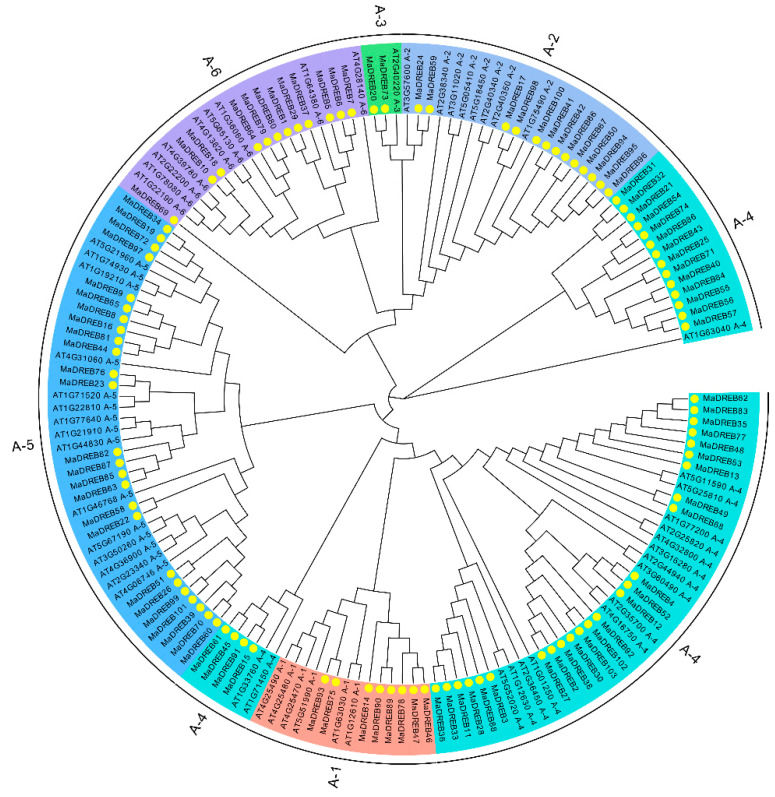
Construction of a tree of the banana *DREB* gene family. The tree illustrates the relationships within the *DREB* family across banana and *Arabidopsis thaliana*.

**Figure 2 plants-13-02119-f002:**
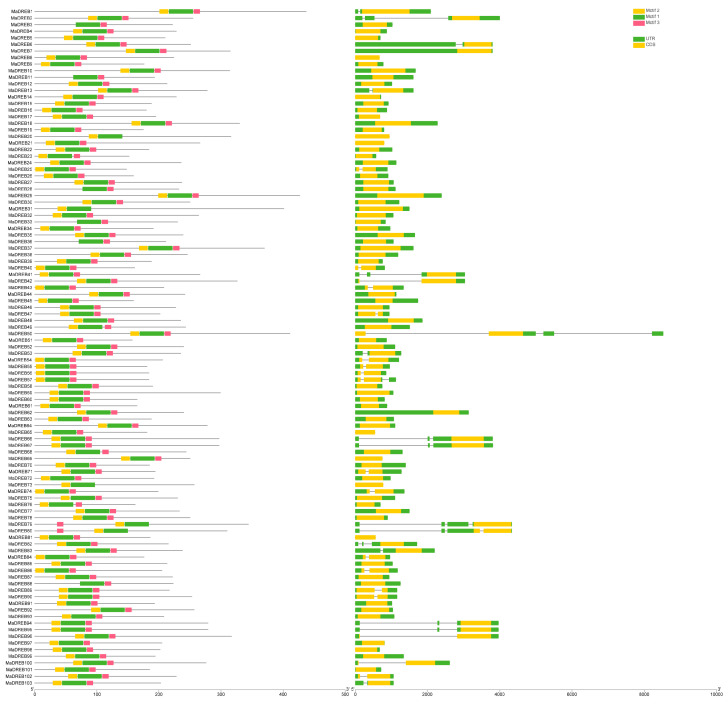
Analysis of DREB gene family domain in banana. The motifs are depicted through the use of three distinct color-coded boxes. A schematic representation of the UTR, exon, and intron regions, where green frames denote the UTR, exons are symbolized by yellow boxes, introns are signified by black lines, providing a clear visualization of the gene structures.

**Figure 3 plants-13-02119-f003:**
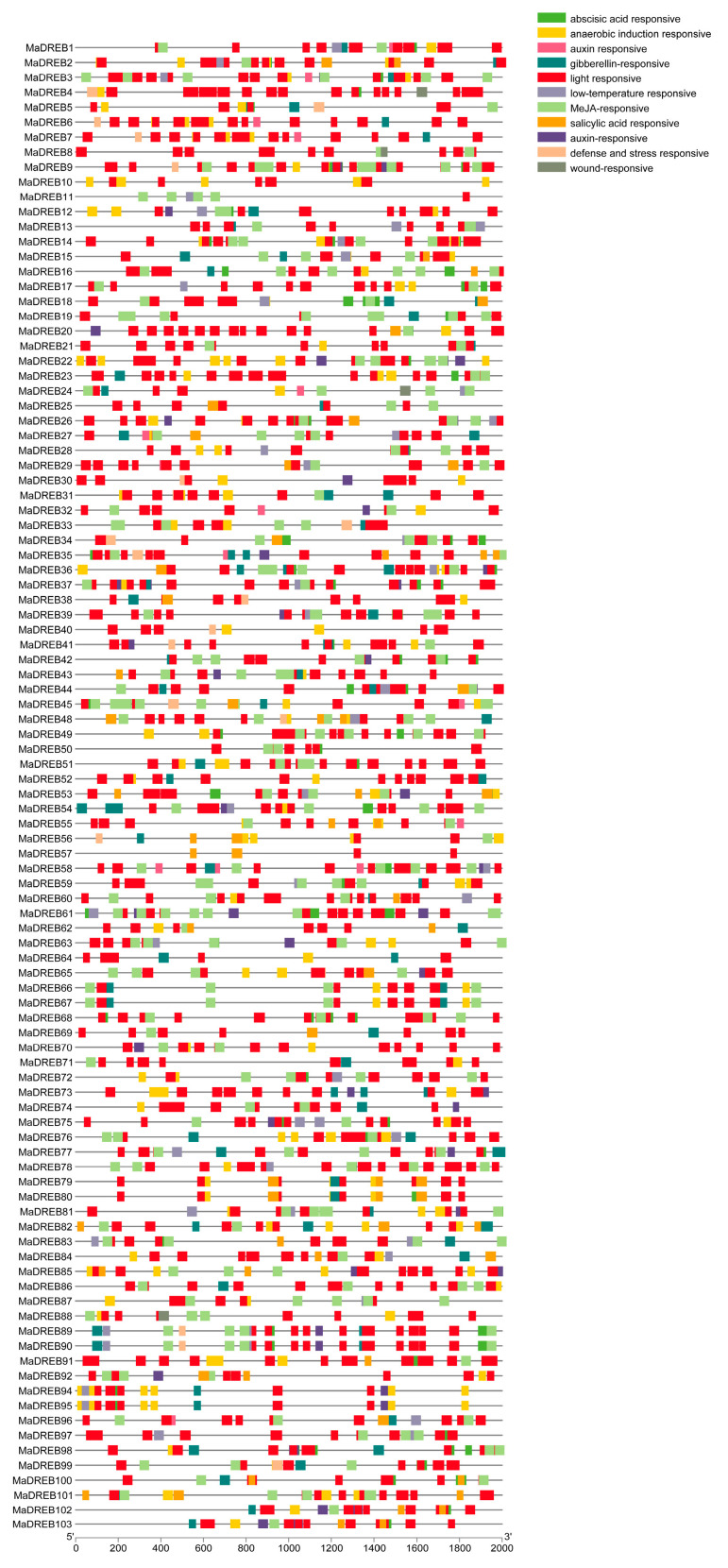
The promoter element analysis. The distribution and density of cis-acting elements within the upstream promoter region of *MaDREBs* are shown.

**Figure 4 plants-13-02119-f004:**
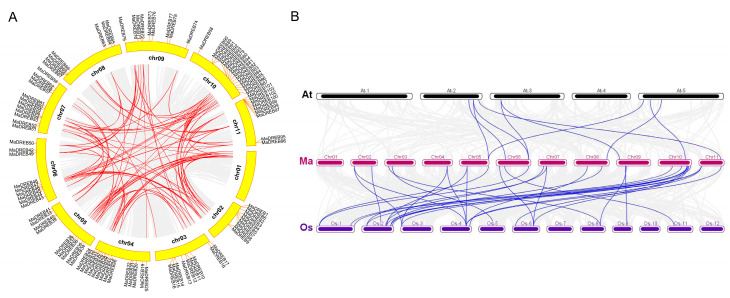
The collinearity distribution of *MaDREBs*. (**A**) Synteny collinearity of banana, Arabidopsis and rice genomes (**B**). The blue line represents the associated gene pairs; in this map, blue lines denote homologous associations between the two species.

**Figure 5 plants-13-02119-f005:**
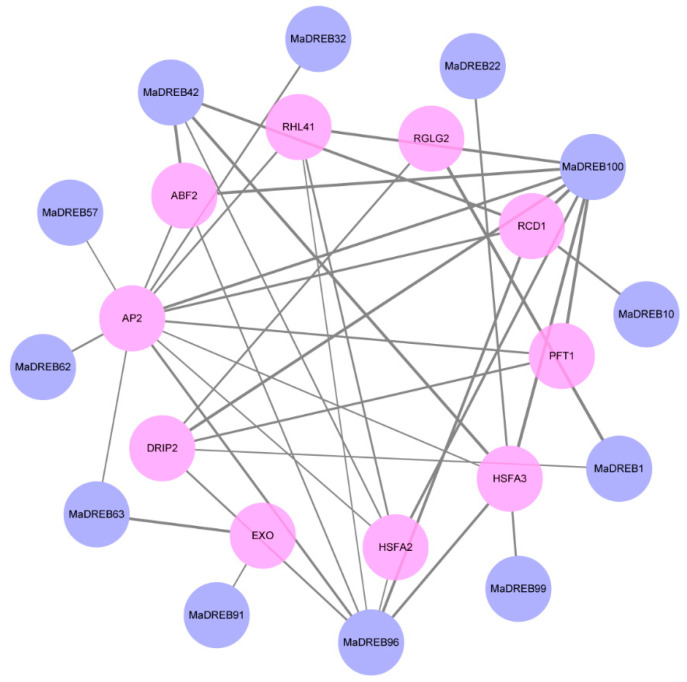
The predicted interaction networks of MaDREB proteins. This visualization provides insights into the potential functional associations and networks that *MaDREB* genes may be a part of within the cellular context.

**Figure 6 plants-13-02119-f006:**
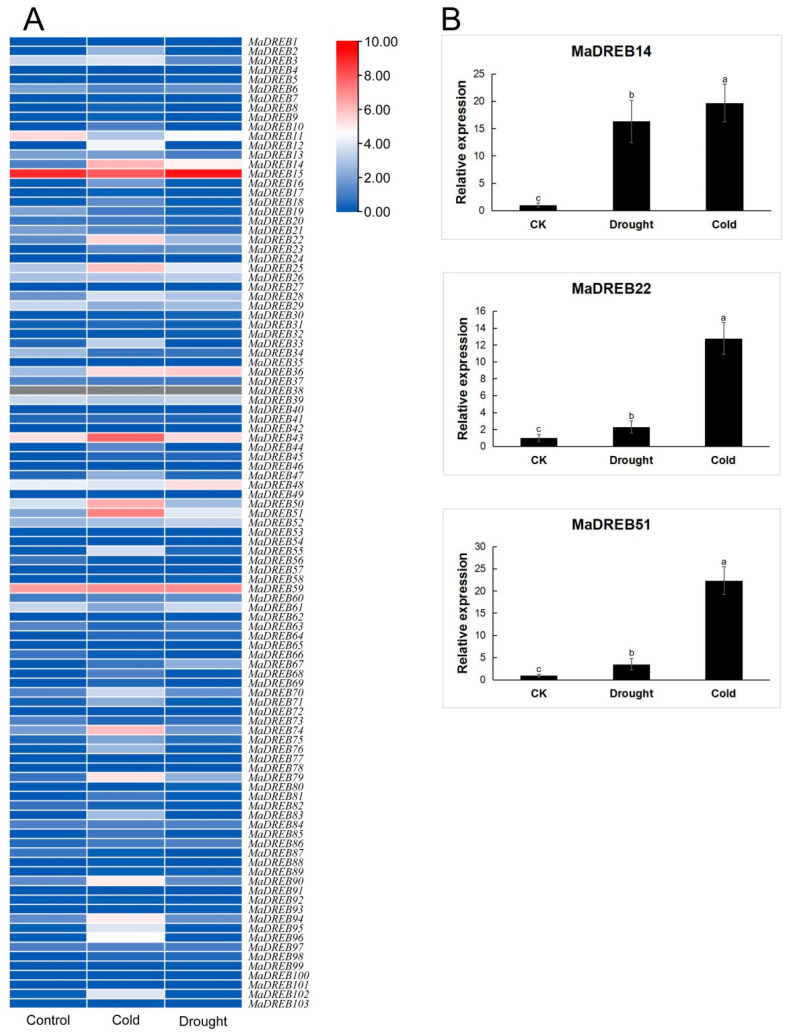
Transcriptome analysis of *MaDREBs* responding to drought and cold stress (**A**) and qRT-PCR of *MaDREBs* responding to drought and cold stress (**B**). Red and blue indicated high and low expression levels, respectively. The a–c was marked on top of the blank bar, which represented differential significance.

**Figure 7 plants-13-02119-f007:**
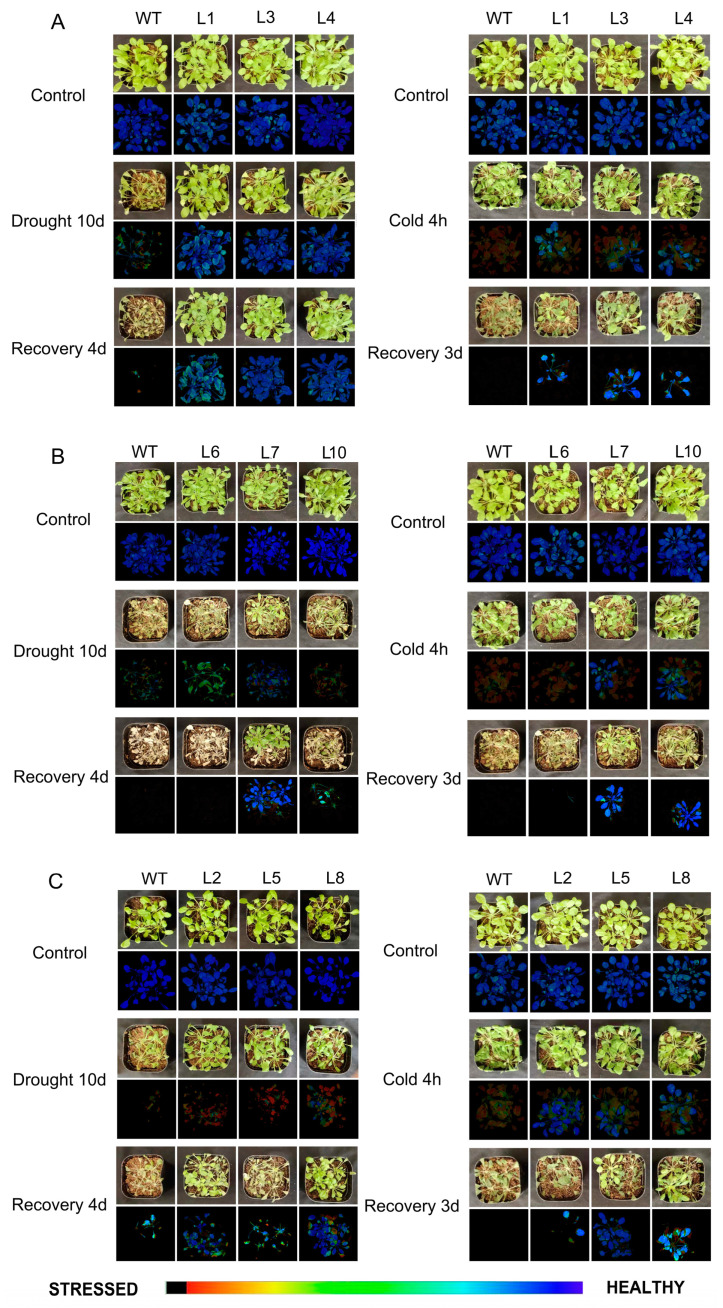
Phenotypes of transgenic Arabidopsis under drought and cold stress treatments and chlorophyll fluorescence measurements. (**A**) *MaDREB14*, (**B**) *MaDREB22*, (**C**) *MaDREB51*.

**Figure 8 plants-13-02119-f008:**
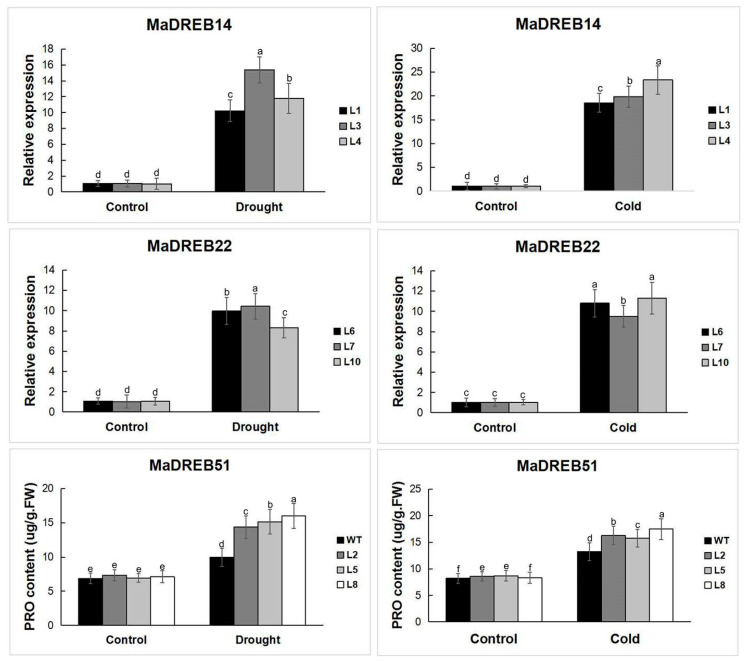
Detection of relative expression of *MaDREB14/22/51* in transgenic Arabidopsis thaliana under drought and cold stress. Data are means ± SD of n = 3 biological replicates. Means denoted by the same letter are not significantly different at *p* < 0.05 as determined by Duncan’s multiple range test.

**Figure 9 plants-13-02119-f009:**
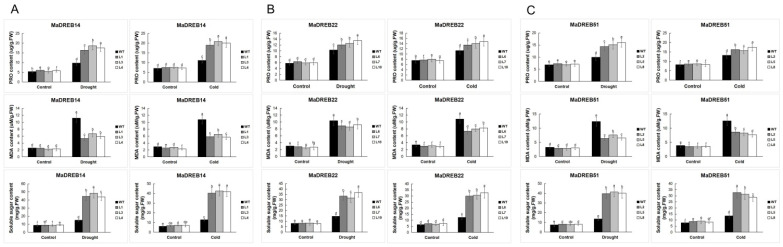
Measurement of physiological indicators in transgenic Arabidopsis thaliana ((**A**) *MaDREB14*; (**B**) *MaDREB22*; (**C**) *MaDREB51*) under drought and cold stress. Data are means ± SD of n = 3 biological replicates. Means denoted by the same letter are not significantly different at *p* < 0.05 as determined by Duncan’s multiple range test.

## Data Availability

The original contributions presented in the study are included in the article/Appendix A, further inquiries can be directed to the corresponding authors.

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
