# Peer review of "Identification of DREB Family Genes in Banana and Their Function under Drought and Cold Stress"

_plants, 2024, doi:10.3390/plants13152119_

Round 1

Reviewer 1 Report

Comments and Suggestions for Authors

The article is a genome-wide analysis of the DREB gene family, which is well known for its role in abiotic stress tolerance. However, there are many problems in the paper, like the English is not up to the mark and needs improvement, and the introduction needs to be a little bit more lengthy so the readers can easily grasp the paper. I have also provided other comments below for the improvement of the paper.

Comment 1: Lines 15-16: The lines needs to be replaced as the first lines of introduction are almost similar.

Comment 2: Line 17: Provide full form for the “DREB” here at its first use.

Comment 3: Line 20: Change “transcriptome sequencing” to “transcriptomic analysis”.    

Comment 4: Lines 21-22: Change the line as “The expression of MaDREB14/22/51 was induced by drought and cold stress, which were further selected for analysis.

Comment 5: Lines 23-24: Please discuss the results in more details, that how the genes confers tolerance against drought and cold. Briefly explain by which mechanism.

Comment 6: Line 29: Change the Keywords to “ Genome-wide, DREB, Climate Change, Drought, Cold.

Comment 7: Provide more details about impact of drought and cold stress on banana. At least 3-4 lines combined to first paragraph of introduction.

Comment 8: Lines 32-34: Change the lines as “Abiotic stresses such as drought and cold temperature limits plant growth and development under climate change circumstances”.

Comment 9: Line 36: Provide scientific name for banana.

Comment 10: Lines 42-50: The lines are providing details for transcription factors, however the information is very general, so just keep it to 2-3 lines and focus more on DREB, provide more details that about DREB role in other horticultural crops.

Comment 11: Line 72: Change the heading to “Phylogenetic analysis”.

Comment 13:  Line 126: Protein interaction analysis: This need to be explained in materials and methods, provide full methodology for the analysis.

Comment 14: Figures: The figures legends are incomplete only one liner is provided, so please explain all the figure legends in details, keep in mind that the figure legend must be self explainable, so the readers can easily understand without going into the results again and again.

Comment 15: Figure 7 should be combined with figure 6.

Comment 16: Figure 9 is not clear, please make it bigger so can be easily seen. And provide a, b, c, about the figures and also explain the details in legends. Separate the gene expressions and physiological parameters and make two figures.

Comments on the Quality of English Language

Author Response

Thank you very much for reviewing the manuscript and pointing out the errors. We have made the relevant changes in the manuscript, the response are listed below. We have also made a number of changes to the language of the manuscript, please check.

Comments 1: Lines 15-16: The lines needs to be replaced as the first lines of introduction are almost similar.

Response 1:The sentence has been changed.

Comments 2:Line 17: Provide full form for the “DREB” here at its first use.

Response 2:The full form of the DREB (dehydration responsive element binding protein)has been added.

Comments 3:Line 20: Change “transcriptome sequencing” to “transcriptomic analysis”.    

Response 3:It has been changed.

Comments 4:Lines 21-22: Change the line as “The expression of MaDREB14/22/51 was induced by drought and cold stress, which were further selected for analysis.

Response 4:It has been changed.

Comments 5:Lines 23-24: Please discuss the results in more details, that how the genes confers tolerance against drought and cold. Briefly explain by which mechanism.

Response 5:Thank you for your suggestion. The sentence has been added as “Additionally, transgenic Arabidopsis plants overexpressing MaDREB14/22/51 exhibited enhanced resistance to drought and cold stress by reducing MDA content and increasing PRO and soluble sugar content.”

Comments 6: Line 29: Change the Keywords to “ Genome-wide, DREB, Climate Change, Drought, Cold.

Response 6:The keywords had been changed.

Comments 7:Provide more details about impact of drought and cold stress on banana. At least 3-4 lines combined to first paragraph of introduction.

Response 7:Thank you for your suggestion. Relevant sentences have been added to the first paragraph of the introduction.

Comments 8:Lines 32-34: Change the lines as “Abiotic stresses such as drought and cold temperature limits plant growth and development under climate change circumstances”.

Response 8:It has been changed.

Comments 9:Line 36: Provide scientific name for banana.

Response 9:The scientific name for banana had been added in the manuscript.

Comments 10:Lines 42-50: The lines are providing details for transcription factors, however the information is very general, so just keep it to 2-3 lines and focus more on DREB, provide more details that about DREB role in other horticultural crops.

Response 10:Thank you for your suggestion. Relevant sentences have been added to the second paragraph of the introduction.

Comments 11:Line 72: Change the heading to “Phylogenetic analysis”.

Response 11:The heading had been changed.

Comments 12:Line 126: Protein interaction analysis: This need to be explained in materials and methods, provide full methodology for the analysis.

Response 12:Thank you for your suggestion. The method of “Protein interaction analysis” has been cited in the literature(Song et al., 2022). The specific methods are as follows: First, the orthovenn2 tool was used (https://orthovenn2.bioinfotoolkits.net/home) to identify the orthologous pairs between MaDREBs and AtDREBs. Second, the interaction networks in which MaDREBs were involved were identified based on the orthologous genes between the banana and Arabidopsis using the AraNetV2 (http://www.inetbio.org/aranet/). The STRING (http://string-db.org/cgi) database and the predicted interaction network were displayed using Cytoscape software (https://cytoscape.org/).

Comments 13:Figures: The figures legends are incomplete only one liner is provided, so please explain all the figure legends in details, keep in mind that the figure legend must be self explainable, so the readers can easily understand without going into the results again and again.

Response 13:Thank you for your suggestion. The figures legends have been added in details. Please check and point it out if need to make any further changes.

Comments 14:Figure 7 should be combined with figure 6.

Response 14:Figures 6 and 7 have been combined together.

Comments 15:Figure 9 is not clear, please make it bigger so can be easily seen. And provide a, b, c, about the figures and also explain the details in legends. Separate the gene expressions and physiological parameters and make two figures.

Response 15:Thank you for your suggestion. The figure had been separated and make two figures in Fig.8 and Fig.9.

Reviewer 2 Report

Comments and Suggestions for Authors

Comments

Comments and Suggestions for Authors

Dear Author,

It is my pleasure to review the manuscript entitled “Identification of DREB family genes in banana and the functional under drought and cold stress” a research article submitted to MDPI Journal, Plants. Authors of this manuscript characterized stress responsive DREB family genes in banana and identified their function in drought and cold stress. They found 103 members of this gene in banana genome. Further characterized three members through expression analysis and found responsive to stresses. Further confirmed their results through overexpressed transgenic Arabidopsis with MaDREB14/22/51. From these results, they concluded that, DREB family genes are regulators of cold and drought tolerance in banana and lays foundation for stress tolerance breeding in banana. The overall experiments, they performed, are well and the results are very convincing and important for banana cultivation. Thus, the presented results take up an important topic consistent with the profile of the Journal.

I have some suggestions, which might improve the manuscript to make important to the wider readers.

·         Improvement in English is necessary for clear understanding

·         Introduction should be more constructive with rationale of the study. Elaborate clearly, why this research is necessary.

·         Include some more related recent work similar to your work even in other plants

·         Demonstrate their lacking and limitations

·         Please check the original PDF where I made comments

·         Similarity percentage should be less than 20%

·         Why only three of the MaDREBs named MaDREB14, MaDREB22, and 67 MaDREB51 have been chosen for further study?

Comments on the Quality of English Language

Extensive editing of English language required

Author Response

Thank you very much for reviewing the manuscript and pointing out the errors. We have made the relevant changes in the manuscript, the response are listed below. Please check.

Comments 1: Improvement in English is necessary for clear understanding

Response 1: Thank you for your suggestion. The entire manuscript has been subjected to very detailed linguistic changes and is presented in revision mode, so please check it and point out if there are any more changes that are needed.

Comments 2: Introduction should be more constructive with rationale of the study. Elaborate clearly, why this research is necessary.

Response 2: Thank you for your suggestion. More detail has been added to the Introduction section, please check.

Comments 3: Include some more related recent work similar to your work even in other plants

Response 3: Relevant similar work has been cited in manuscripts. For example, Song et al., 2022; Xu et al., 2023a,b; Zhang et al., 2023; Ma et al.,2024.

Comments 4: Demonstrate their lacking and limitations

Response 4: We added a future outlook for the study, add the last sentence in the conclusion section.

Comments 5: Please check the original PDF where I made comments

Response 5: The relevant changes have been shown in the manuscript, please check. Please indicate if there are any further changes that need to be made.

Comments 6: Similarity percentage should be less than 20%

Response 6:  A number of language changes and rewrites have been made in the manuscript. The Similarity percentage is less than 20%.

Comments 7: Why only three of the MaDREBs named MaDREB14, MaDREB22, and 67 MaDREB51have been chosen for further study?

Response 7: In this manuscript, a total of 103 banana DREB genes were identified and some of the DREB members were able to respond to drought and low temperature stress. Among them, three members, MaDREB14, MaDREB22,MaDREB51, were able to respond to both drought and low-temperature stresses and had the highest expression levels. Therefore, the above three members were selected for further experiments.

Round 2

Reviewer 2 Report

Comments and Suggestions for Authors

Article has been improved substantially. However, still need further correction in English as well as content. Please see the pdf.

Comments on the Quality of English Language

English need further correction

Author Response

Thank you very much for reviewing the manuscript again. We have corrected the errors marked in the manuscript accordingly, as well as checking the English language in detail and making corrections, if there are still inappropriate places, please help us point out! The corresponding changes are as follows, Please check:

Comments 1:”the functional” of the title changed into “their function”.

Response 1: The change has been made.

Comments 2: Change the sentence ”The industry is affected by drought and low temperature stress.”

Response 2: The sentence has been changed into “Drought and low temperature stress affect the growth of banana”.

Comments 3:Fig.3 still not with high resolution.

Response 3:As the image is too large to be placed in the manuscript and will appear blurred, you can consult the original image in the document, which is very clear.

Comments 4: Follow format for Plants at “Materials and methods” section.

Response 4: The “Materials and methods” has been formatted according to the format of plants

Comments 5: Change the sentence “Moreover, some research has shown that DREBs play a role crucial role in regulating gene expression in response to abiotic stressors such as drought, cold, and high-intensity stresses.”

Response 5: The sentence has been changed into “In addition, several studies have shown that DREBs play a crucial role in regulating gene expression in response to abiotic stresses such as drought and cold.”

Comments 6: How old are the seedlings used for the experiment?

Response 6: The seedlings used for the experiment are at five-leaf stage.